# Unbiased Prototype Consistency Learning for Multi-Modal and Multi-Task Object Re-Identification

**Zhongao Zhou**[1*]  **Bin Yang**[1*]  **Wenke Huang**[1]   **Jun Chen**[1†]  **Mang Ye**[1†]
[1] School of Computer Science, Wuhan University

## Abstract

In object re-identification (ReID) task, both cross-modal and multi-modal retrieval methods have achieved notable progress. However, existing approaches are designed for specific modality and category (person or vehicle) retrieval task, lacking generalizability to others. Acquiring multiple task-specific models would result in wasteful allocation of both training and deployment resources. To address the practical requirements for unified retrieval, we introduce Multi-Modal and Multi-Task object ReID ($M^3$T-ReID). The $M^3$T-ReID task aims to utilize a unified model to simultaneously achieve retrieval tasks across different modalities and different categories. Specifically, to tackle the challenges of modality distibution divergence and category semantics discrepancy posed in $M^3$T-ReID, we design a novel Unbiased Prototype Consistency Learning (UPCL) framework, which consists of two main modules: Unbiased Prototypes-guided Modality Enhancement (UPME) and Cluster Prototype Consistency Regularization (CPCR). UPME leverages modality-unbiased prototypes to simultaneously enhance cross-modal shared features and multi-modal fused features. Additionally, CPCR regulates discriminative semantics learning with category-consistent information through prototypes clustering. Under the collaborative operation of these two modules, our model can simultaneously learn robust cross-modal shared feature and multi-modal fused feature spaces, while also exhibiting strong category-discriminative capabilities. Extensive experiments on multi-modal datasets RGBNT201 and RGBNT100 demonstrates our UPCL framework showcasing exceptional performance for $M^3$T-ReID. The code is available at https://github.com/ZhouZhongao/UPCL.

## 1   Introduction

Object re-identification (ReID) [75, 49, 65, 45, 18, 60, 3, 9, 75, 73, 69, 30, 4] leverages computer vision techniques to identify specific objects (such as persons or vehicles) in videos and still images. ReID technology has been widely applied in intelligent video surveillance, public security, and other related fields. Traditional ReID predominantly focuses on single-modal scenario, where both the query and gallery consist of RGB images. However, RGB cameras are highly sensitive to illumination variations, making it difficult to accurately capture target information under low-light or overexposed conditions. To address the above challenges, Near-Infrared (NI) and Thermal Infrared (TI) modalities have been introduced into ReID tasks, enabling robust imaging in challenging environments [72, 15, 29, 55, 17, 71, 6, 63]. Depending on the retrieval scenarios, the existing ReID methods can be broadly categorized into cross-modal ReID [58, 53, 36, 70, 41, 19, 2] and multi-modal ReID [47, 50, 68, 67]. Specifically, cross-modal ReID focuses on retrieval between two different modalities (e.g.,NI-RGB, TI-RGB), whereas multi-modal ReID utilizes RGB, NI, and TI fusion to achieve feature matching.

---

[*]Equal contribution.
[†]Corresponding Author.

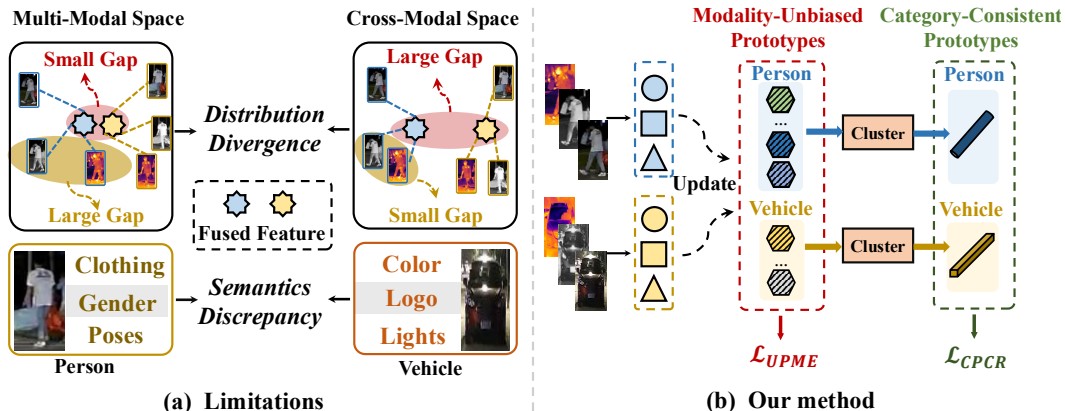

Figure 1: **Illustration of limitations in M³T-ReID and our method.** (a) *Distribution divergence*: In multi-modal feature space, the multi-modal fused features exhibit smaller gap while cross-modal gap remain larger. Conversely, the opposite characteristic holds in cross-modal feature space. *Semantics discrepancy*: Different categories of objects possess distinct discriminative semantics. (b) We introduce modality-unbiased prototypes and cluster-derived category-consistent prototypes to enhance the model's comprehensive retrieval capability from both modality and category perspectives.

As illustrated in Figure 1, although existing cross-modal and multi-modal approaches have achieved remarkable results, they still suffer from two significant limitations: **1)** Real-world surveillance environments present extreme complexity, where targets may appear in scenarios captured by either single-modality cameras or aligned multi-modal imaging systems. While cross-modal ReID primarily focuses on learning a shared cross-modal feature space, multi-modal ReID emphasizes the effective fusion of different modalities to obtain more robust fused features. The cross-modal and multi-modal approaches follow different optimization directions, thereby resulting in distribution divergence. Therefore, existing ReID models cannot simultaneously handle both cross-modal and multi-modal retrieval. **2)** The retrieval tasks are confined to a specific category, necessitating separate model training for either person or vehicle ReID tasks. In practical scenarios such as criminal investigations, surveillance systems require the capability to simultaneously retrieve both suspects and vehicles. Due to the discrepancy in semantics between diverse categories, existing ReID methods lack the generalization capability to perform unified retrieval across categories. While training dedicated retrieval models for distinct modalities and categories may serve as a feasible solution, this approach inevitably leads to substantial redundancy in both training and deployment resources.

To meet the real-world demands for retrieval across diverse modalities and categories, we propose the Multi-Modal and Multi-Task object ReID (M³T-ReID). The M³T-ReID task aims to achieve a unified model for simultaneous retrieval across multiple modalities and diverse categories. However, achieving high-performance of M³T-ReID introduces several challenges. Firstly, since cross-modal and multi-modal retrieval models optimize fundamentally different objectives, this leads to **challenge I** : *How to jointly learn both a robust cross-modal shared feature space and an effective multi-modal fusion feature space.* Secondly, discriminative features vary significantly across object categories. For instance, person ReID primarily focuses on attributes like pose and clothing while vehicle ReID emphasizes vehicle type and color, which raises the **challenge II** : *How to enable a model to simultaneously learn category-specific discriminative representations for heterogeneous objects.*

To address the aforementioned chanllenges in M³T-ReID, we propose Unbiased Prototype Consistency Learning framework (UPCL) which comprises two key modules: Unbiased Prototypes-guided Modality Enhancement (UPME) and Cluster Prototype Consistency Regularization (CPCR). For **challenge I**, UPME enhances both cross-modal shared features and multi-modal fused features through modality-unbiased prototypes, thereby bridging the discrepancy across heterogeneous modalities and simultaneously improving robustness of the overall feature space. For **challenge II**, CPCR derives category-consistent features through prototypes clustering, thereby regulating the model to stably acquire category-specific discriminative semantics from diverse categories.

The main contributions of this paper can be summarized as follows:

- To address the practical demands for retrieval of diverse modalities and categories, we propose a novel Multi-Modal and Multi-Task object ReID ($M^3$T-ReID).
- To address the challenges in $M^3$T-ReID, we propose UPCL which comprises two main components: UPME and CPCR. UPME leverages modality-unbiased prototypes to simultaneously enhance cross-modal shared features and multi-modal fused features, and CPCR regulates the learning of discriminative semantics across diverse object categories through prototypes clustering.
- Extensive experiments on the public multi-modal ReID benchmarks RGBNT201 and RGBNT100 have verified the advantage of our methods, achieving significantly higher accuracy compared to existing counterparts in both cross-modal and multi-modal retrieval scenarios.

## 2 Related Work

### 2.1 Cross-modal and Multi-modal Re-identification

**Cross-modal re-identification** [23, 14, 20, 5, 8, 62, 31, 59, 11] aims to retrieve target RGB images across heterogeneous modalities. The cross-modal retrieval capability of a model primarily depends on the robustness of itscross-modal shared feature space. Wu *et al.*[52] utilize a zero-padding one-stream network with grayscale inputs to learn the shared feature between RGB andNI images. Ye *et al.*[61] introduce a Channel Augmentation (CA) mechanism that mitigates the modality gap by generating color-irrelevant person representations. Liu *et al.*[23] propose the Memory-Augmented Unidirectional Metric (MAUM) method to enhance the cross-modality correlation by utilizing two unidirectional metrics. Liang *et al.*[20] make an early attempt at unsupervised cross-modal ReID with a two-stage framework. Yang *et al.*[57] design an Augmented Dual-Contrastive Aggregation (ADCA) learning framework for Unsupervised Learning Visible-Infrared Person ReID.

**Multi-modal re-identification** [47, 15] jointly leverages complementary information from multiple modalities to extract more robust fused features, thereby improving retrieval accuracy. Zheng *et al.*[72] propose PFNet which hierarchically fuses RGB,NI, and TI features to obtain more robust representations. Wang *et al.*[49] design a Cross-Modal Interacting Module to enhance modality-specific information during feature fusion. Wang *et al.*[50] utilize the relationship among heterogeneous modalities to fine-tune the network prior to inference, thereby improving generalization to unseen data. Zhang *et al.*[68] propose a general PromptMA framework, which employs learnable prompts to aggregate modalities and bridge the modality distribution gap. Wang *et al.*[46] introduce the token permutation to enhance inter-modal interaction and facilitate multi-spectral feature alignment. Zhang *et al.*[67] propose EDITOR framework, which obtains spatial–frequency masks to refine multi-modal features. Wang *et al.*[48] adaptively balances decoupled features using a mixture-of-experts mechanism to produce more robust multi-modal representations.

Due to the divergent optimization objectives between cross-modal and multi-modal, a single model cannot be effectively applied to both retrieval scenarios simultaneously. Furthermore, significant semantic discrepancies exist among objects of different categories, models trained on one category cannot achieve generalized to retrieval of other categories. To address diverse retrieval requirements in real-world scenarios, we propose the UPCL framework, which trains a unified model capable of performing retrieval across multiple modalities and object categories.

### 2.2 Prototypes Learning

Prototype is calculated as the mean feature of the instances belonging to the same ID [38]. Due to its simplicity and scalability, prototype plays an indispensable role across various domains, such as few-shot learning [56, 38, 13, 27, 42] , unsupervised learning [54, 7, 16, 53],incremental learning [74, 76, 66, 43, 37], and federated learning [21, 12, 26, 22, 28, 25, 40, 64, 34, 35]. In object ReID research [24, 57, 58, 53], prototypes and other feature-centric concepts have also demonstrated significant effectiveness. Luo *et al.*[24] introduce Center Loss to enhance the feature similarity of same-ID samples in the model. Yang *et al.*[57] assigns pseudo-labels to unannotated data through clustering and dynamically update the centers of samples corresponding to each pseudo-ID as supervisory signals for network optimization.

Current applications of prototypes or similar concepts primarily leverage their statistical properties at ID-level to enhance feature compactness within the same category, thereby improving network robustness. In this work, we innovatively utilize modality-unbiased prototypes from a modality-

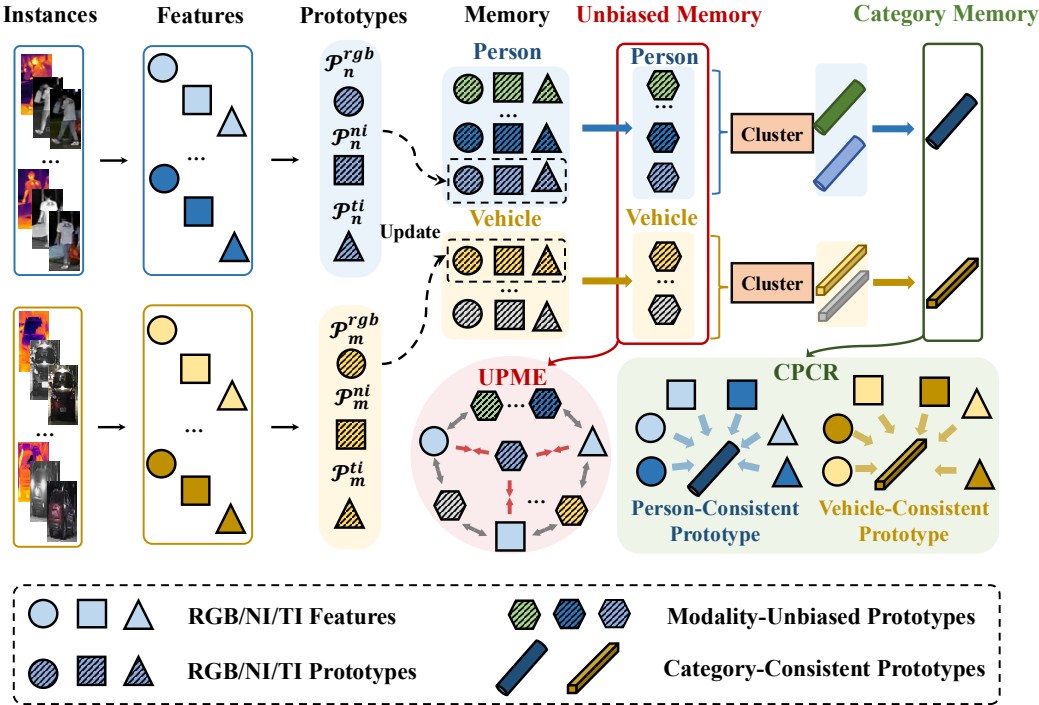

Figure 2: **Overview of of the proposed UPCL framework.** It consists of two core components: Unbiased Prototypes-guided Modality Enhancement (UPME) and Cluster Prototype Consistency Regularization (CPCR). We dynamically update the multi-modal prototype memory for each ID during every training iteration. UPME aggregates modality-unbiased prototypes via Equation (9), leveraging their identity-consistent information to strengthen both cross-modal and multi-modal representation learning. CPCR further utilizes category-consistent prototypes clustered from modality prototypes, exploiting their category-consistent semantics to regularize the model and achieve more discriminative category-wise decision boundaries.

consistency perspective and employ clustering strategies to obtain category-level (rather than identity-level) discriminative features. This approach significantly enhances the unified ReID model's capability to perform robust retrieval across multiple categories and modalities.

## 3  Method

In this section, we present the Unbiased Prototype Consistency Learning framework (UPCL) which consists of two main modules. The Unbiased Prototypes-guided Modality Enhancement (UPME) module leverages modality-unbiased prototypes to simultaneously enhance both cross-modal shared features and multi-modal fused features, thereby improving the model's performance across different retrieval modes. The Cluster Prototype Consistency Regularization (CPCR) module utilizes modality-unbiased prototypes via clustering to derive category-consistent prototypes, which are then utilized to regulate the model's discriminative semantic learning process for different categories. The overview of UPCL is illustrated in Figure 2 and the details are discussed in the following subsections.

### 3.1  Overall Architecture

Our method utilizes the pretrained CLIP [32] model as the visual encoder which is shared with RGB, NI and TI modalities. Specifically, for the $i$-th multi-modal instance $V_i = \{V_i^{rgb}, V_i^{ni}, V_i^{ti}\}$, the images of three modalities are cropped into equal-sized patches and then mapped to embedding vectors with fixed dimensions. Next, we feed these embedding vectors into the visual encoder to obtain their corresponding class token $f_i^m \in \mathbb{R}^D$ and patch tokens $p_i^m \in \mathbb{R}^{N_p \times D}$, where $m \in \{rgb, ni, ti\}$, $N_p$ denotes the number of patch tokens and $D$ is the embedding dimension.

Consistent with existing multi-modal methods [72, 49, 50, 46], we employ the label smoothing cross-entropy loss $\mathcal{L}_{ce}$ [39] and triplet loss $\mathcal{L}_{tri}$ [10] to supervise the learning of visual encoder:

$$\mathcal{L}_g = \mathcal{L}_{ce} + \mathcal{L}_{tri}. \tag{1}$$

To further encourage cross-modal feature alignment, we introduce an additional cross-modal alignment loss function:

$$\mathcal{L}^{(m \to n)}(i) = -\log \frac{\exp\left(\langle f_i^m, f_i^n \rangle / \tau\right)}{\sum_{j=1}^{B} \exp\left(\langle f_i^m, f_j^n \rangle / \tau\right)}, \tag{2}$$

$$\mathcal{L}^{(n \to m)}(i) = -\log \frac{\exp\left(\langle f_i^n, f_i^m \rangle / \tau\right)}{\sum_{j=1}^{B} \exp\left(\langle f_i^n, f_j^m \rangle / \tau\right)}, \tag{3}$$

where $\langle \cdot, \cdot \rangle$ represents the cosine similarity function. $f_i^m$ and $f_i^n$ denote the features of $i$-th instance with different modalities $m, n \in \{rgb, ni, ti\}$. $B$ is the batch size, and $\tau$ is the temperature parameter. Then the cross-modal loss function between modalities $m$ and $n$ can be formulated as:

$$\mathcal{L}_{m \leftrightarrow n} = \frac{1}{2B} \sum_{i=1}^{B} [\mathcal{L}^{(m \to n)}(i) + \mathcal{L}^{(n \to m)}(i)]. \tag{4}$$

Building upon this, we derive the cross-modal loss function $\mathcal{L}_{cross}$:

$$\mathcal{L}_{cross} = \mathcal{L}_{rgb \leftrightarrow ni} + \mathcal{L}_{rgb \leftrightarrow ti} + \mathcal{L}_{ni \leftrightarrow ti}. \tag{5}$$

Finally, we obtain the base objective $\mathcal{L}_b$ of the framework:

$$\mathcal{L}_b = \mathcal{L}_g + \mathcal{L}_{cross}. \tag{6}$$

## 3.2 Unbiased Prototypes-guided Modality Enhancement

To enable the model to achieve high performance in both cross-modal and multi-modal retrieval, it is essential to design optimization strategies that maintain directional consistency between these two paradigms. Inspired by the successful application of prototypes in other domains, we leverage the identity-consistent information encapsulated in prototypes to guide the feature learning.

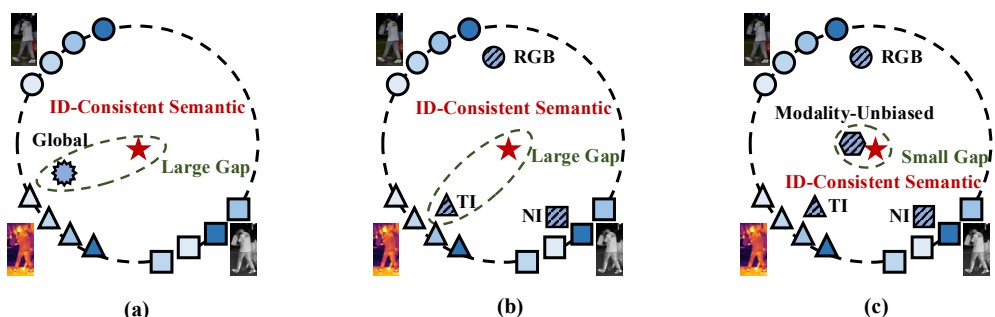

Figure 3: Comparison of different prototypes. (a) Global prototypes; (b) Modality-specific prototypes; (C) Modality-unbiased Prototypes (Ours). It is obviously observed that our proposed Modality-Unbiased Prototypes exhibit superior ID-consistent semantic representation in the feature space.

We compare two commonly used prototypes (global prototypes and modality-specific prototypes) in multi-modal learning alongside our newly proposed unbiased prototype in Figure 3.

**Global Prototypes**. The features of all samples across three modalities under the same ID are fused to obtain the global prototype:

$$\mathcal{P}_c^{global} = \frac{1}{\|I(c)\|} \sum_{i \in I(c)} L_{fused}(f_i^{rgb}, f_i^{ni}, f_i^{ti}), \tag{7}$$

where $I(c)$ denotes the indices of all instances with identity $c$. $L_{fused}$ is an MLP designed to fuse the features from the three modalities, producing a global feature in $\mathbb{R}^D$.

**Modality-Specific Prototypes**. We compute modality-specific prototypes for identity $c$ :

$$\mathcal{P}_c^m = \frac{1}{\|I(c)\|} \sum_{i \in I(c)} f_i^m, m \in \{rgb, ni, ti\}. \tag{8}$$

Then this yields a set of prototypes $\mathcal{P}_c = \{\mathcal{P}_c^{rgb}, \mathcal{P}_c^{ni}, \mathcal{P}_c^{ti}\}$ for three modalities .

**Modality-Unbiased Prototypes**. Global prototypes tend to incorporate semantic features biased toward the dominant modality, thereby deviating from identity-consistent semantics. Conversely, modality-specific prototypes primarily focus on intra-modal identity information, which inherently carries modality-specific bias relative to identity consistency. Therefore, in order to optimize the features towards identity-consistent semantics, we comprehensively derive a modality-unbiased prototype from $\mathcal{P}_c = \{\mathcal{P}_c^{rgb}, \mathcal{P}_c^{ni}, \mathcal{P}_c^{ti}\}$:

$$\mathcal{U}_c = (\mathcal{P}_c^{rgb} + \mathcal{P}_c^{ni} + \mathcal{P}_c^{ti})/3. \tag{9}$$

As illustrated in Figure 3, compared with global prototypes and modality-specific prototypes, the modality-unbiased prototypes mitigate both inter-modal and intra-modal distribution discrepancies, thereby thereby capturing more robust identity representations.

Under the guidance of modality-unbiased prototypes, we enhance features of modality $m$ by:

$$\mathcal{L}_{UPME}^m = -\frac{1}{B} \sum_{i=1}^{B} \log \frac{\exp\left(\langle f_i^m, \mathcal{U}_{c_i} \rangle / \tau\right)}{\sum_{j=1}^{B} \exp\left(\langle f_i^m, \mathcal{U}_{c_j} \rangle / \tau\right)}, \tag{10}$$

where $c_i$ represents the identity of the $i$-th sample. Finally, we formulate the Unbiased Prototypes-guided Modality Enhancement loss through summation:

$$\mathcal{L}_{UPME} = \mathcal{L}_{UPME}^{rgb} + \mathcal{L}_{UPME}^{ni} + \mathcal{L}_{UPME}^{ti}. \tag{11}$$

By enhancing semantic consistency across multi-modal features, the discriminative capability for identities in any modality is improved, thereby boosting the cross-modal retrieval performance. Simultaneously, the approach reduces inter-modal divergence and improves fusion efficiency, yielding more robust multi-modal representations.

### 3.3 Cluster Prototype Consistency Regularization

Current ReID methods typically train dedicated models for specific object categories (e.g., persons or vehicles). However, since discriminative semantic features vary significantly across different object categories, models trained on specific category lack generalization capability. To develop a unified model capable of retrieving diverse object categories, we need to enable the model to accurately capture category-consistent semantic information. As detailed in Section 3.2, the modality-unbiased prototypes inherently encapsulate identity-consistent semantic information that is strongly correlated with sample categories. This intrinsic correlation motivates us to explore a novel approach for extracting category-consistent semantics directly from these prototypes.

Assuming that there are $N_p$ and $N_v$ identities belonging to persons and vehicles, respectively, we utilize $\mathcal{U}^p = \{\mathcal{U}_c\}_{c=1}^{N_p}$ and $\mathcal{U}^v = \{\mathcal{U}_c\}_{c=N_p+1}^{N_p+N_v}$ to denote the sets of modality-unbiased prototypes. To obtain category-consistent information, we aim to fully integrate prototypes from all identities within a specific category. While the identities of any two prototypes in $\mathcal{U}^p$ or $\mathcal{U}^h$ are distinct, they may share highly similar semantic characteristics (e.g. males wearing short sleeves). Such semantic similarity naturally forms clusters among these identities, where the dominant clusters contribute more representative information within $\mathcal{U}^p$ or $\mathcal{U}^v$.

To better capture discriminative semantic features of specific categories, we propose to utilize cluster-level statistics instead of identity-level information. The acquisition of reliable cluster-level statistics fundamentally depends on achieving proper clustering of $\mathcal{U}^p$ and $\mathcal{U}^v$. Unlike conventional clustering algorithms K-means [1] and HAC [51], FINCH [33] operates in a completely parameter-free manner, automatically determining the optimal number of clusters based on the inherent similarity relationships among all prototypes. Therefore, we employ FINCH to cluster $\mathcal{P}^{u,p}$ and $\mathcal{P}^{u,v}$, obtaining the sets of multiple clustered prototypes $\mathcal{C}^p$ and $\mathcal{C}^v$:

$$\mathcal{U}^p = \{\mathcal{U}_c\}_{c=1}^{N_p} \xrightarrow{Cluster} \mathcal{C}^p = \{\mathcal{C}_l^p\}_{l=1}^{L_p}, \tag{12}$$

$$\mathcal{U}^v = \{\mathcal{U}_c\}_{c=N_p+1}^{N_v} \xrightarrow{Cluster} \mathcal{C}^v = \{\mathcal{C}_l^v\}_{l=1}^{L_v}, \tag{13}$$

where $L_p$ and $L_v$ denote the number of elements in $\mathcal{C}^p$ and $\mathcal{C}^v$. The $l$-th cluster prototypes for person and vehicle are denoted as $\mathcal{C}_l^u$ and $\mathcal{C}_l^u$, which are calculated by averaging all modality-unbiased prototypes in $l$-th cluster. Then we obtain the category-consistent prototype through:

$$\mathcal{C}ate^p = \frac{1}{L_p} \sum_{l=1}^{L_p} \mathcal{C}_l^p, \tag{14}$$

$$\mathcal{C}ate^v = \frac{1}{L_v} \sum_{l=1}^{L_v} \mathcal{C}_l^v. \tag{15}$$

Then, we introduce the CPCR loss function with $\mathcal{C}ate^p$ and $\mathcal{C}ate^v$ for specific category:

$$\mathcal{L}_{CPCR}^p = \frac{1}{\|I^p\|} \sum_m \sum_{i \in I^p}^{B^p} \log \frac{\exp\left(\langle f_i^m, \mathcal{C}ate^p \rangle / \tau\right)}{\exp\left(\langle f_i^m, \mathcal{C}ate^p \rangle / \tau\right) + \exp\left(\langle f_i^m, \mathcal{C}ate^v \rangle / \tau\right)}, \tag{16}$$

$$\mathcal{L}_{CPCR}^v = \frac{1}{\|I^v\|} \sum_m \sum_{i \in I^v}^{B^v} \log \frac{\exp\left(\langle f_i^m, \mathcal{C}ate^v \rangle / \tau\right)}{\exp\left(\langle f_i^m, \mathcal{C}ate^p \rangle / \tau\right) + \exp\left(\langle f_i^m, \mathcal{C}ate^v \rangle / \tau\right)}, \tag{17}$$

where $I^p$ and $I^v$ denote the index sets of all person-class and vehicle-class instances within a batch, respectively. It is natural to derive the total CPCR loss:

$$\mathcal{L}_{CPCR} = \mathcal{L}_{CPCR}^p + \mathcal{L}_{CPCR}^v. \tag{18}$$

Under the regularization of category-consistent prototypes, the model effectively learns discriminative semantic features that represent object categories, thereby achieving robust performance across diverse retrieval tasks involving different object categories.

Finally, we integrate all components to formulate a comprehensive optimization objective:

$$\mathcal{L}_{total} = \mathcal{L}_b + \alpha \mathcal{L}_{UPME} + \beta \mathcal{L}_{CPCR}, \tag{19}$$

where $\alpha$ and $\beta$ are the the hyper-parameters to balance the contributions of each loss function.

## 4  Experiments

### 4.1  Experimental Settings

**Datasets and Evaluation Protocols.** To evaluate the performance of our UPCL, we combined RGBNT201 [72] and RGBNT100 [15] to construct a multi-modal dataset with diverse object categories. Specifically, RGBNT201 is the first multi-modal person ReID dataset, where each pedestrian ID contains RGB, NI and TI modalities. RGBNT100 contains the same modalities as RGBNT201, but collects vehicle images instead. We evaluate the performance of our proposed UPCL with Rank-$k$ matching accuracy, mean Average Precision (mAP) which are the commonly utilized metrics in object ReID tasks. The evaluation protocol encompasses six cross-modal testing scenarios ($R \to N$, $N \to R$, $R \to T$, $T \to R$, $N \to T$, and $T \to N$) along with one multi-modal testing configuration (*RNT* $\to$ *RNT*).

**Implementation Details.** The implementation platform is Pytorch with a NVIDIA 3090 GPU. We utilize the pre-trained CLIP as the visual encoder. Images of all modalities are resized to 256×128. For data augmentation, we apply random horizontal flipping, cropping and erasing. The batch size is set to 64, sampling 8 images per identity. The training process is conducted with the Adam optimizer for 50 epochs, and the initial learning rate is set to 3.5e-4. We select 0.03 as the temperature parameter $\tau$. The hyper-parameters $\alpha$ and $\beta$ are set as 2.0 and 0.5 respectively. During the testing phase, cross-modal retrieval directly computes similarity using features from two modalities, while multi-modal retrieval concatenates features from three modalities for matching.

Table 1: Comparison with the state-of-the-art methods. Each model is trained on a combined dataset consisting of RGBNT201 and RGBNT100, and evaluated separately.

| Methods | RGBNT201 | | | | | | | | | | | | | |
| | $R \to N$ | | $N \to R$ | | $R \to T$ | | $T \to R$ | | $N \to T$ | | $T \to N$ | | $RNT \to RNT$ | | Harm_Mean | |
| | mAP | Rank-1 | mAP | Rank-1 | mAP | Rank-1 | mAP | Rank-1 | mAP | Rank-1 | mAP | Rank-1 | mAP | Rank-1 | mAP | Rank-1 |
|---|---|---|---|---|---|---|---|---|---|---|---|---|---|---|---|---|
| HTT [50] | 4.43 | 3.31 | 3.59 | 3.26 | 2.73 | 0.48 | 3.26 | 0.36 | 3.26 | 2.27 | 4.25 | 4.31 | 9.16 | 4.67 | 3.83 | 1.05 |
| TOP-ReID [46] | 10.50 | 8.61 | 10.71 | 9.33 | 3.55 | 1.32 | 3.54 | 1.20 | 5.40 | 4.31 | 6.10 | 4.67 | 63.74 | 64.95 | 6.27 | 3.07 |
| PromptMA [68] | 20.37 | 17.70 | 19.39 | 14.35 | 11.80 | 7.06 | 13.77 | 12.32 | 11.56 | 8.49 | 10.28 | 6.34 | 65.71 | 68.18 | 15.32 | 10.95 |
| EDITOR [67] | 3.70 | 2.03 | 3.38 | 2.27 | 3.83 | 2.39 | 3.72 | 0.60 | 4.37 | 2.51 | 3.32 | 0.96 | 56.03 | 56.70 | 4.26 | 1.56 |
| DeMo [48] | 3.98 | 2.27 | 4.33 | 2.63 | 3.30 | 1.44 | 4.09 | 3.59 | 3.10 | 0.60 | 3.44 | 2.75 | 64.35 | 63.76 | 4.22 | 1.82 |
| UPCL (Ours) | **22.33** | **23.21** | **20.09** | **18.54** | **16.77** | **14.59** | **18.19** | **18.54** | **17.93** | **14.47** | **17.56** | **21.17** | 64.91 | 67.12 | **20.75** | **19.97** |

| Methods | RGBNT100 | | | | | | | | | | | | | |
| | $R \to N$ | | $N \to R$ | | $R \to T$ | | $T \to R$ | | $N \to T$ | | $T \to N$ | | $RNT \to RNT$ | | Harm_Mean | |
| | mAP | Rank-1 | mAP | Rank-1 | mAP | Rank-1 | mAP | Rank-1 | mAP | Rank-1 | mAP | Rank-1 | mAP | Rank-1 | mAP | Rank-1 |
|---|---|---|---|---|---|---|---|---|---|---|---|---|---|---|---|---|
| HTT [50] | 5.94 | 5.89 | 4.64 | 2.33 | 3.00 | 0.64 | 3.59 | 1.05 | 3.57 | 2.33 | 3.90 | 2.33 | 32.21 | 53.00 | 4.48 | 1.76 |
| TOP-ReID [46] | 15.47 | 20.06 | 12.24 | 12.01 | 3.50 | 2.86 | 3.17 | 1.40 | 3.96 | 4.31 | 3.80 | 1.92 | 71.47 | 89.15 | 5.48 | 3.57 |
| PromptMA [68] | 41.73 | 53.64 | 42.64 | 51.78 | 8.54 | 7.46 | 9.96 | 8.28 | 7.87 | 4.90 | 10.37 | 8.92 | 71.07 | 86.53 | 13.93 | 11.28 |
| EDITOR [67] | 2.59 | 1.40 | 2.99 | 2.04 | 3.45 | 2.92 | 3.20 | 2.39 | 2.61 | 1.57 | 3.19 | 2.16 | 74.25 | 93.00 | 3.44 | 2.28 |
| DeMo [48] | 3.12 | 1.69 | 4.14 | 2.39 | 2.42 | 0.76 | 3.07 | 2.33 | 4.31 | 2.51 | 4.55 | 3.27 | 79.12 | 93.47 | 3.97 | 2.02 |
| UPCL (Ours) | **48.41** | **64.08** | **49.83** | **64.61** | **16.50** | **19.42** | **17.20** | **18.31** | **17.26** | **21.40** | **17.25** | **18.08** | **79.41** | **94.87** | **22.63** | **27.50** |

## 4.2 Comparison with State-of-the-art Methods

We present a comprehensive comparison of UPCL with state-of-the-art methods as outlined in Table 1. To thoroughly evaluate the model's holistic performance on six cross-modal and one multi-modal testing scenarios, we employ the harmonic mean of task-specific metrics as the aggregated performance measure, given that the harmonic mean places greater emphasis on smaller values compared to the arithmetic mean.

As demonstrated in Table 1, on the person-category dataset (RGBNT201) , UPCL significantly outperforms all competing methods across all six cross-modal scenarios, which strongly validates its cross-modal matching capability. For multi-modal retrieval results, compared with PromptMA which contains module specifically designed for multi-modal fusion matching, our approach trails by merely 0.8 percentage points in mAP. Regarding the arithmetic mean of all seven metrics, UPCL achieves substantial superiority over other methods, surpassing the second-best approach by 5.43 and 9.02 percentage points on these two metrics, respectively. For the vehicle-category dataset (RGBNT100) , our UPCL demonstrates consistent superiority across every scenario. Across both category-specific datasets (person and vehicle) under all seven evaluation protocols, our method achieves remarkable performance, which conclusively demonstrates its effectiveness for the M$^3$T-ReID task.

## 4.3 Ablation Studies

To thoroughly validate the effectiveness of our proposed UPME and CPCR modules, we conducted extensive ablation experiments on both RGBNT201 and RGBNT100 datasets. The experimental protocol involves incrementally integrating our proposed modules into the baseline model without introducing any extra modifications, enabling direct performance comparison through controlled ablation studies. As illustrated in Table 2, all reported metrics are presented as the harmonic mean of performance across seven evaluation scenarios.

**Effect of UPME.** By leveraging the identity-consistent information captured in modality-unbiased prototypes, the UPME module enhances feature representations across different modalities, thereby boosting cross-modal and multi-modal representation capabilities. The experimental results demonstrate that our UPME module achieves mAP improvements of 3.20% and 2.19% on RGBNT201 and RGBNT100 respectively, compared to the baseline. When aggregated across both datasets, the module delivers average gains of 2.41% mAP and 3.37% rank-1 accuracy.

**Effect of CPCR.** The CPCR module clusters modality-unbiased prototypes from UPME to derive category-consistent prototypes, then exploits their category-discriminative semantics to guide model optimization. After incorporating CPCR, the model achieves a 4.10% mAP and 5.99% rank-1 improvement on RGBNT201, while attaining 0.73% mAP and 2.91% rank-1 gains on RGBNT100.

Table 2: Ablation study of each component on RGBNT201 and RGBNT100.

| Components | | RGBNT201 | | | | RGBNT100 | | | | Average | | | |
|---|---|---|---|---|---|---|---|---|---|---|---|---|---|
| UPME | CPCR | mAP | Rank-1 | Rank-5 | Rank-10 | mAP | Rank-1 | Rank-5 | Rank-10 | mAP | Rank-1 | Rank-5 | Rank-10 |
| | | 13.85 | 10.06 | 20.62 | 27.93 | 19.89 | 21.78 | 24.39 | 26.39 | 16.87 | 15.92 | 22.51 | 27.16 |
| ✔ | | 16.65 | 13.98 | 25.59 | 33.56 | 21.90 | 24.59 | 28.06 | 30.61 | 19.28 | 19.29 | 26.83 | 32.09 |
| ✔ | ✔ | **20.75** | **19.97** | **36.03** | **46.13** | **22.63** | **27.50** | **31.13** | **32.79** | **21.69** | **23.74** | **33.58** | **39.46** |

**Multi-modal Feature Space**      **Cross-modal Feature Space**

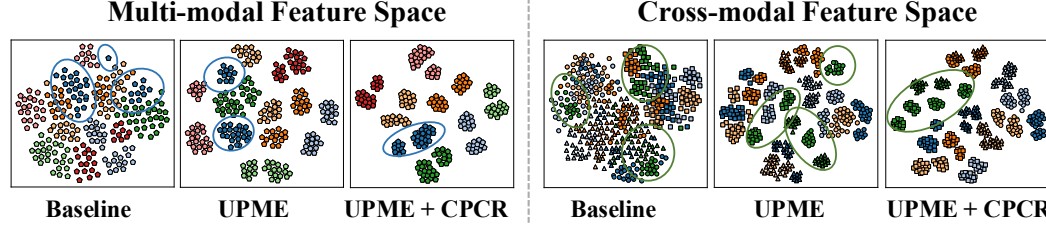

**Baseline**    **UPME**    **UPME + CPCR**     **Baseline**    **UPME**    **UPME + CPCR**

Figure 4: The t-SNE visualization of several randomly selected identities. Color indicate identities.

**Visualization.** To intuitively demonstrate the effectiveness of UPME and CPCR, we visualize the t-SNE[44] feature distribution of several identities in Figure 4. Specifically, the three subplots on the left visualize the embedding space of multi-modal fused features. Under the effects of UPME and CPCR, the fused features of the same ID progressively converge, while the discriminability between different identities becomes increasingly pronounced. The cross-modal features of identical IDs in the right figure exhibit consistent variation trends. Therefore, the t-SNE visualization clearly indicates that integrating UPME and CPCR enhances the intra-identity consistency and inter-identity discriminability of both fused and cross-modal features.

**Effects of Diverse Categories** . As discussed earlier, different categories of retrieval targets contain distinct discriminative semantics, making it challenging for a model to maintain high performance across retrieval tasks involving diverse categories. To validate this claim, Table 3 presents a comparison between category-specific model and a unified model. The category-specific model is trained on a single-category dataset, either RGBNT201 or RGBNT100, while the unified model is trained on a multi-category dataset formed by combining RGBNT201 and RGBNT100. It can be

Table 3: Comparison between specifc and unified model. Each model is trained on a combined dataset consisting of RGBNT201 and RGBNT100, and evaluated separately. mAP(%) is reported.

| Methods | RGBNT201 | | RGBNT100 | |
|---|---|---|---|---|
| | Specific | Unified | Specific | Unified |
| HTT [50] | 69.0 | 9.16 | 75.7 | 32.21 |
| TOP-ReID [46] | 72.3 | 63.74 | 81.2 | 71.47 |
| PromptMA [68] | 78.4 | 65.71 | 85.3 | 71.07 |
| EDITOR [67] | 66.5 | 56.03 | 79.8 | 74.25 |
| DeMo [48] | 79.0 | 64.35 | 86.2 | 79.12 |

clearly observed that, for the same method, the unified model exhibits a significant performance drop compared to its corresponding category-specific model. This suggests that mixing multiple categories in training introduces substantial interference, hindering the model's ability to simultaneously learn discriminative semantics for all categories. These findings strongly support our hypothesis that multi-category training can negatively impact model optimization.

## 5 Cross-domain Retrieval Evaluation

In conventional Re-ID evaluations, both training and testing sets are typically collected under the same scene conditions. As a result, the learned feature space may be constrained to a specific environment, making the evaluation results insufficient to reflect the generalization capability required in real-world applications. To assess the cross-domain generalization ability of our model, we train it on a combined dataset of RGBNT201 and RGBNT100, and evaluate it on the unseen MSVR310 dataset.

The comparison results are presented in Table 4. All models show significantly lower detection accuracy on the MSVR310 dataset compared to RGBNT201 and RGBNT100, indicating that cross-domain generalization remains highly challenging for existing Re-ID models. In particular, methods

Table 4: Performance analysis of cross-domain generalization on MSVR310 dataset. All models are trained on the combined dataset of RGBNT201 and RGBNT100, and evaluated on the MSVR310 dataset. Rank-$k$ accuracy (%) and mAP(%) are reported.

| Methods | MSVR310 | | | | | | | | | | | | | |
| | $R \rightarrow N$ | | $N \rightarrow R$ | | $R \rightarrow T$ | | $T \rightarrow R$ | | $N \rightarrow T$ | | $T \rightarrow N$ | | $RNT \rightarrow RNT$ | | $Harm\_Mean$ | |
| | mAP | Rank-1 | mAP | Rank-1 | mAP | Rank-1 | mAP | Rank-1 | mAP | Rank-1 | mAP | Rank-1 | mAP | Rank-1 | mAP | Rank-1 |
|---|---|---|---|---|---|---|---|---|---|---|---|---|---|---|---|---|
| HTT [50] | 1.99 | 1.18 | 2.00 | 1.52 | 1.88 | 1.18 | 2.33 | 1.34 | 1.91 | 1.02 | 1.88 | 2.20 | 5.90 | 8.97 | 2.20 | 1.02 |
| TOP-ReID [46] | 4.31 | 6.43 | 3.87 | 3.21 | 2.23 | 0.85 | 1.84 | 1.52 | 2.30 | 1.02 | 2.54 | 2.71 | 14.91 | 25.55 | 2.94 | 1.89 |
| PromptMA [68] | 11.92 | 14.72 | 10.71 | 13.37 | 2.43 | 1.02 | 2.64 | 1.52 | 2.82 | 2.53 | 3.70 | 3.21 | 23.14 | 31.98 | 4.28 | 2.78 |
| EDITOR [67] | 2.16 | 1.18 | 1.53 | 0.52 | 1.59 | 0.51 | 1.49 | 0.63 | 1.45 | 0.85 | 1.35 | 1.02 | 17.05 | 30.96 | 1.70 | 0.82 |
| DeMo [48] | 2.12 | 0.34 | 1.73 | 0.17 | 1.87 | 1.35 | 1.77 | 1.02 | 1.66 | 1.02 | 1.76 | 0.51 | 15.89 | 26.90 | 2.07 | 0.52 |
| UPCL (Ours) | 13.89 | 20.30 | 14.09 | 20.14 | 9.43 | 10.15 | 9.03 | 9.14 | 9.57 | 11.68 | 9.35 | 11.17 | 24.60 | 38.07 | 11.44 | 13.77 |

such as HTT, TOP-ReID, EDITOR, and DeMo exhibit severe performance degradation under both cross-modal and multi-modal testing scenarios on the unseen MSVR310 dataset. In contrast, the UPCL and PromptMA methods achieve considerably better results, with UPCL consistently outperforming PromptMA across various detection settings. These experimental results provide strong evidence that our model possesses excellent cross-domain generalization capability. The superior cross-domain retrieval performance of UPCL further validates our method from another perspective—not only does it enhance the robustness of the fused feature space across modalities, but it also strengthens the model's ability to discriminate semantic features across different categories.

## 6 Limitations

Although our method has demonstrated promising performance on existing multi-modal ReID datasets, it still exhibits several notable limitations. The current multi-modal ReID datasets are relatively scarce, mainly restricted to the pedestrian and vehicle domains. Considering the practical demands of real-world applications, it is crucial to explore how incorporating a broader range of categories affects retrieval performance. In addition, our unified model may slightly underperform task-specific models in a very limited number of test scenarios. Although this is reasonable, this also indicates that our approach still has potential for further enhancement. Therefore, our future work will focus on extending UPCL to more generalized multi-modal and multi-task retrieval scenarios.

## 7 Conclusion

In this paper, we introduce a novel M³T-ReID task to address the practical demands of retrieval across diverse modalities and categories. To tackle the challenges in M³T-ReID, we propose the Unbiased Prototype Consistency Learning framework (UPCL) which consists of two main modules UPME and CPCR. Specifically, UPME mitigates the divergence between cross-modal shared spaces and multi-modal fusion distributions by leveraging identity-consistent information from modality-unbiased prototypes, thereby enhancing both cross-modal and multi-modal representations. Meanwhile, CPCR reduces semantic discrepancies across categories by clustering modality-unbiased prototypes to obtain category-consistent prototypes with discriminative semantics. Extensive experiments on multiple datasets validate the superiority and effectiveness of our method, demonstrating its robustness and generalization ability in diverse retrieval scenarios.

## Acknowledgements

This work is partially supported by National Natural Science Foundation of China under Grants (62501428, 62176188), the Innovative Research Group Project of Hubei Province under Grants (2024AFA017), the Major Project of Science and Technology Innovation of Hubei Province (2024BCA003, 2025BEA002), Postdoctoral Fellowship Program of China Postdoctoral Science Foundation (GZC20241268, 2024M762479), Hubei Postdoctoral Talent Introduction Program (2024HBB-HJD070), Hubei Provincial Natural Science Foundation of China (2025AFB219) and WHU-Kingsoft Joint Lab. The numerical calculations in this paper had been supported by the super-computing system in the Supercomputing Center of Wuhan University.

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
