# OpenReview forum: "Unbiased Prototype Consistency Learning for Multi-Modal and Multi-Task Object Re-Identification"
_NeurIPS.cc/2025/Conference — NeurIPS 2025 spotlight_

### Official Review · Reviewer_NDQB · 2025-06-26

**Clarity:** 4
**Significance:** 3
**Originality:** 4
**Rating:** 5
**Confidence:** 4

**Summary:**

This paper introduces a new and practically important Re-Identification (Re-ID) scenario with both heterogeneous modalities and different object categories. To address the challenges posed by this setting, the authors propose UPCL to hierarchically leverage the modality-agnostic and category-specific cues embedded in prototypes. Extensive experiments on two public multi-modal ReID datasets verify the effectiveness of UPCL.

**Questions:**

See the weaknesses above.

**Ethical Concerns:**

["NO or VERY MINOR ethics concerns only"]

**Final Justification:**

The proposed method is novel, to the best of my knowledge, this paper is the first to analyze how heterogeneous modalities and diverse object categories impact Re-ID models, and it presents a concise yet effective approach that boosts performance across a range of retrieval scenarios. During the rebuttle, the ahthous have well responsed my concerns. Besides, three reviewers consistently tends to accept it and one give the borderline reject. Based on above consideration, I tend to accept it.

**Limitations:**

yes

**Paper Formatting Concerns:**

No Formatting Concerns

**Quality:**

3

**Strengths And Weaknesses:**

Strengths：

1. To the best of my knowledge, this paper is the first to analyze how heterogeneous modalities and diverse object categories impact Re-ID models, and it presents a concise yet effective approach that boosts performance across a range of retrieval scenarios.
2.The manuscript is is well-structured with a coherent narrative flow, and the accompanying figures are clear and informative.
3. To tackle retrieval challenges stemming from heterogeneous modalities and different categories, the motivation and formulation of the core modules are articulated concisely：（1）Unbiased Prototypes-guided Modality Enhancement (UPME) simultaneously enhances cross-modal representations and multimodal fusion features. （2）Cluster Prototypes Consistent Regularization (CPCR) regularizes the learning of discriminative semantics over diverse object categories.
4. Extensive experiments show that the approach achieves superior performance across multiple evaluation settings on the RGBNT201 and RGBNT100 benchmarks.
5. In cross-domain comparison experiments, the proposed method consistently achieves the best generalization performance.

Weakness：

1. Multimodal Re-ID benchmarks usually resize vehicle and pedestrian images to category-specific resolutions to suit their respective retrieval scenarios. In this study, however, images from all categories are uniformly normalized to 256 × 128 during experimentation. Might this protocol diverge from real-world deployment conditions or incur adverse effects?
2. Are the temperature hyperparameters in Eqs. (10) and (16) instantiated with the same value? Moreover, the manuscript does not examine how varying the temperature affects model performance, so an ablation study on this hyperparameter is strongly recommended.
3. In each training epoch, are the prototypes recomputed from scratch, kept static, or updated through a momentum-based mechanism? The paper does not explicitly specify the prototype update mechanism.
4. The module label in Figure 4 contains a typographical error: “UMPE” should be revised to “UPME.”

---

> ### Author Rebuttal · Authors · 2025-07-31
>
> ### **Response to Reviewer NDQB**
>
> Thank you for your detailed review and helpful suggestions. Your feedback is highly appreciated and has played an important role in enhancing the presentation and technical depth of our work. We will release our code to provide the implementation details.
>
> ---
> ### Weaknesses & Questions
> **W 1) :  Effect of Image Resizing on Performance**
>
> **A1** : We normalize images from different categories to a resolution of 256×128 and use a shared tokenizer for processing. In theory, employing separate tokenizers for pedestrians and vehicles while sharing a common encoder could potentially lead to better performance. However, the imbalance in data distribution across different categories may introduce optimization difficulties. Furthermore, our experimental results reveal that resizing vehicle images to the same resolution as pedestrian images has a relatively minor effect on retrieval performance. Specifically, we conduct experiments with DeMo on the RGBNT100 dataset with different input resolutions (256×128 and 128×256). The results of mAP/R1 under 256×128 and 128×256 are 86.0/97.2 and 86.7/97.7 . There exists only a negligible performance difference of around 0.7%/0.5% , which can be attributed to the fact that most original images have widths smaller than 128 pixels, resulting in minimal data compression or loss during resizing.
>
>
> **W 2) :  Analysis of Temperature Hyperparameter**
>
> **A2** : The temperature hyperparameters used in Equations (10) and (16) are set to the same value. We have supplemented ablation studies on the temperature hyperparameter. The detailed experimental results are shown in the table below. As observed, our method demonstrates its effectiveness across different temperature settings.
>
> *Table. **Ablation study of temperature hyperparameters $\tau$
>
> | $\alpha$|  RGBNT201  | RGBNT100  | Average |
> | ------  |  --- | --- | --- |
> |         | mAp ｜Rank-1| mAp ｜Rank-1|mAp ｜Rank-1|
> |   0.01   | 19.45｜18.29|  20.44｜23.23| 19.95｜20.76
> |   **0.03**|   **20.75**｜**19.97**|  **22.63**｜**27.50**| **21.69**｜**23.74**
> |   0.05   |18.49｜17.96|  22.36｜25.41| 20.43｜21.54
> |   0.07   |  18.89｜17.68|  21.98｜24.72| 20.44｜21.20
> |   0.09   |  18.30｜17.58|  20.87｜23.55| 19.59｜20.57
>
> **W 3) :  Computation Details of Prototypes**
>
> **A3** :  At the beginning of each training epoch, we recalculate the prototype corresponding to each identity, which is then used in the subsequent training process. Thank you for pointing out this omission, we will provide a detailed explanation of this computation procedure in the manuscript.
>
>
> **W 4) :  Textual Errors**
>
> **A4** :Thank you for your suggestion. We will revise and correct the textual errors in the  the final version.

---

> > ### Comment · Reviewer_NDQB · 2025-08-03
> >
> > The authors have well responsed my concerns, I keep my positive score.

---

> > > ### Author Response · Authors · 2025-08-08
> > >
> > > Dear Reviewer NDQB,
> > >
> > > We sincerely thank you for your positive evaluation of our work. We greatly value your insightful feedback and the effort you invested in reviewing our paper. Your suggestions have provided clear directions for improvement, and we will incorporate them in the revised version.
> > >
> > > All the best,
> > >
> > > Authors.

---

### Official Review · Reviewer_U2xk · 2025-06-26

**Clarity:** 4
**Significance:** 4
**Originality:** 3
**Rating:** 5
**Confidence:** 5

**Summary:**

This paper introduces a novel Multi-Modal and Multi-Task object Re-Identification (M3T-ReID) problem, aiming to deploy a unified model that simultaneously handles person and vehicle retrieval under both multi-modal and cross-modal settings. To solve it, the authors propose UPCL framework with two modules: UPME builds modality-agnostic prototypes to align the cross-modal shared space as well as the multi-modal fusion space. CPCR clusters these prototypes to impose category-consistent semantics across heterogeneous classes. Extensive experiments on RGBNT201 (person) and RGBNT100 (vehicle) show that UPCL surpasses SOTA multi-modal ReID methods and delivers consistent gains in both multi-modal and cross-modal tasks.

**Questions:**

1. The experimental comparison focuses on multi-modal methods. Have the authors evaluated UPCL against strong uni-modal or traditional cross-modal ReID approaches?
2. The number of identities and the number of images per identity in the RGBNT201 and RGBNT100 datasets are quite different. Did the authors apply any balancing strategy during training or study how this imbalance affects performance?

**Ethical Concerns:**

["NO or VERY MINOR ethics concerns only"]

**Final Justification:**

The response has addressed all my concerns, and I will raise my rating score.

**Limitations:**

Yes

**Quality:**

3

**Strengths And Weaknesses:**

Strengths
1. The manuscript is clearly written, logically structured, and easy to follow.
2. Training a unified Re-ID model that can simultaneously handle heterogeneous modalities and object categories is a highly meaningful goal. M3T-ReID task proposed in this paper carries strong practical significance. Moreover, the framework and ideas presented here can further spur the development of more general-purpose Re-ID.
3. The UPME and CPCR proposed in the paper promote M3T-ReID learning through modality-unbiased prototypes and category prototypes. The framework remains conceptually simple while directly addressing both modality divergence and semantic heterogeneity.
4. The prototype-based representation strategy introduced in this work is highly inspiring and readily transferable to other domains—a commendable innovation.
5. The model is benchmarked on two types of datasets under seven settings (six cross-modal + one multi-modal), demonstrating strong and consistent performance improvements.

Weaknesses
1. This paper only uses one person dataset RGBNT201 and one vehicle dataset RGBNT100 to implement of different types of objects ReID. Incorporating additional categories or environments would better validate the robustness claimed for M3T-ReID.
2. I think pseudocode can be added to show the process of the proposed UPCL framework more clearly. This can lower the barrier for researchers in adjacent domains to leverage and build upon your methodology.
3. In Figure 4 and L269-L276, the UPME module is incorrectly labeled as UMPE or UMPR.
4. All reported experiments are conducted exclusively on the hybrid RGBNT201 + RGBNT100 dataset, with no attempt to evaluate the proposed approach on a mixed RGBNT201 + MSVR310 setting.

---

> ### Author Rebuttal · Authors · 2025-07-31
>
> ### **Response to Reviewer U2xk**
>
> Thank you for your time and thoughtful review. We appreciate your positive feedback on our contributions, writing clarity, and experimental design. We will release our code to provide greater clarity and transparency regarding the implementation details. We hope our responses below address your concerns and lead to a favorable reassessment.
>
> ---
>
> ### Weaknesses & Questions
>
> **Q 1) :  Comparison with Traditional Cross-modal Methods**
>
> **A1** :  The cross-modal method DEEN achieves Rank-1 accuracies of 22.41, 12.77, and 8.78 on R-N, R-T, and N-T on the RGBNT201 dataset, respectively. Under more challenging settings involving multiple retrieval modalities and increased category interference, our method achieves corresponding accuracies of 23.21, 14.59 and 14.47. It can be observed that our method consistently outperforms DEEN all three scenarios.
>
>
> **Q 2): Imbalance in the Number of Identities and Images**
>
> **A2** : We acknowledge that there exists a significant discrepancy in the number of identities and the number of images per identity between the RGBNT201 and RGBNT100 datasets. However, the primary focus of this work is to address the challenges arising from heterogeneous retrieval modalities and category variations, rather than applying any strategy to balance the distribution of identities or image counts. That said, we agree that a unified ReID model should be capable of handling such imbalanced scenarios in real-world applications. Therefore, we plan to explore strategies for mitigating these sample imbalance issues in our future work.
>
> **W 1) & W 4) :  Dataset Selection and Configuration**
>
> **A3** :  1） **Scarcity of Multimodal ReID Datasets**. Your suggestion aligns well with the limitations we discussed in Section 5 of our paper. Indeed, incorporating additional object categories or environmental contexts would further demonstrate the effectiveness and generalizability of our proposed method. However, existing multi-modal ReID datasets are still relatively limited, mostly constrained to pedestrian and vehicle categories. In future work, we will actively monitor the release of new multi-modal datasets covering other categories, and we also plan to construct our own datasets for these categories. These will be integrated into our framework to support more comprehensive and in-depth studies.
>
> 2） **Selection of Datasets for Cross-Domain Generalization**. Given the similarity in data collection environments between RGBNT201 and RGBNT100, we conduct the main experiments in our paper on the combined RGBNT201 + RGBNT100 dataset to validate the effectiveness of our method in both cross-modal and cross-category scenarios. In contrast, MSVR310 is collected under a different setting, and thus we use it separately for cross-domain generalization experiments to evaluate the generalization ability of the model trained on RGBNT201 + RGBNT100.
>
>
> **W 2) & W 3) : Textual Errors and Pseudocode**
>
> **A4** :  Thank you for your suggestion. We will revise  the textual errors and include the corresponding pseudocode in the supplementary materials of the final version to facilitate reproduction and further development by other researchers.

---

> > ### Comment · Area_Chair_2AF1 · 2025-08-07
> > **Discussion**
> >
> > Dear Reviewer U2xk,
> >
> > Thank you for your service to NeruIPS 2025 paper review. What do you think of the author rebuttal? Did they address your concerns? Could you please kindly help share your further opinions? Thank you.
> >
> > Best regards,
> > Your AC

---

### Official Review · Reviewer_aGKs · 2025-06-30

**Clarity:** 4
**Significance:** 4
**Originality:** 4
**Rating:** 5
**Confidence:** 5

**Summary:**

The paper introduces a new Multi-Modal and Multi-Task Re-ID (M3T-ReID) task and proposes the UPCL framework, which couples Unbiased Prototypes-guided Modality Enhancement (UPME) and Cluster Prototypes Consistent Regularization (CPCR). UPME leverages identity-consistent information from modality-unbiased prototypes to simultaneously enhance both cross-modal and multi-modal feature representations. CPCR performs clustering on modality-unbiased prototypes to derive category-consistent prototypes with discriminative semantics. Experiments demonstrate the challenge of the M3T-ReID task and the effectiveness of the UPCL framework.

**Questions:**

(1). The prototype memory bank is rebuilt at the start of every epoch; what is the additional computational and time overhead introduced by this step in practical training?
(2). Have you performed a sensitivity study on the loss-weight hyperparameters to show that the method is robust across a reasonable range of settings?

**Ethical Concerns:**

["NO or VERY MINOR ethics concerns only"]

**Final Justification:**

I have no more concerns. Good luck!

**Limitations:**

Yes

**Quality:**

4

**Strengths And Weaknesses:**

This study introduces a well-motivated M3T-ReID task that explicitly addresses the multimodal and cross-category challenges encountered in real-world Re-ID scenarios. Building on this, the proposed UPCL framework has been exhaustively validated—its constituent modules supported by extensive experiments and compelling empirical evidence demonstrating the method’s overall robustness and high quality.
The task definition and the high-level framework are easy to follow, yet the description of each prototype-learning step could be more explicit. In addition, a more detailed introduction of the dataset can make it easier for readers to understand, and L229 should be sampling eight pairs or instances rather than images.
By widening Re-ID toward a universal paradigm that spans both task types and object categories, M3T-ReID opens a fertile avenue for future unified retrieval research.
While using prototypes is not entirely new, applying modality-unbiased and category-consistent prototype learning in such a broad multi-task setting is a reasonable contribution, though the paper should clarify differences from the most recent prototype-based methods.

---

> ### Author Rebuttal · Authors · 2025-07-31
>
> ### **Response to Reviewer aGKs**
>
> We greatly appreciate the reviewer’s time and effort in evaluating our paper. We are grateful for your thoughtful feedback, which has guided us in refining our manuscript. We will release our code to better illustrate the details.
>
> ---
> ### Weaknesses & Questions
> **Q 1) :  Construction of the Prototype Memory Bank**
>
> **A1** :
>  We recalculate the prototype for each identity at the start of every epoch. Although this introduces additional computational overhead, the prototypes encapsulate both class-level and identity-discriminative information, which significantly enhances retrieval accuracy across various modalities.
>
> **Q 2) : Ablation Study of Loss Weight Hyperparameters**
>
> **A2** :   We conduct ablation studies to evaluate the impact of different loss weight configurations for α and β. The detailed results are presented in the table below. As shown, our proposed method consistently achieves strong and stable performance across a wide range of loss weight settings, demonstrating its robustness to hyperparameter variations.
>
> *Table. **Ablation study of loss-weight hyperparameters $\alpha$ and $\beta$***
>
> | $\alpha$| $\beta$ |  RGBNT201  | RGBNT100  | Average |
> | ------  | ------  | --- | --- | --- |
> |         |         | mAp ｜Rank-1| mAp ｜Rank-1|mAp ｜Rank-1|
> |   0.5   |   0.5   |17.93｜17.16|  21.97｜25.47| 19.95｜21.32
> |   1.0   |   0.5   |19.37｜18.64|  22.11｜26.86| 20.74｜21.85
> |   **2.0**   |   **0.5**  |**20.75**｜**19.97**|  **22.63**｜**27.50**| **21.69**｜**23.74**
> |   2.0   |   1.0   |20.39｜19.75|  22.52｜27.17| 21.46｜23.46
> |   2.0   |   2.0   |20.27｜19.18|  22.23｜27.09| 21.25｜23.14

---

> ### Comment · Reviewer_aGKs · 2025-08-05
>
> Thanks for the author's response. My concerns are well solved. I have also checked other reviewers' comments and found no more concerns. Therefore, I will keep my score.

---

> > ### Author Response · Authors · 2025-08-08
> > **Thank-you Letter**
> >
> > Dear Reviewer aGKs,
> >
> > Thank you very much for your recognition and for raising the evaluation of our work. We truly appreciate your constructive feedback and the time you dedicated to reviewing our paper. Your comments have been invaluable in helping us improve the quality and clarity of our work, and we will revise the paper accordingly.
> >
> > All the best,
> >
> > Authors.

---

### Official Review · Reviewer_wjAM · 2025-07-03

**Clarity:** 2
**Significance:** 2
**Originality:** 3
**Rating:** 4
**Confidence:** 5

**Summary:**

This method proposes to address both multi-modal and cross-modal re-id tasks, and person and vehicle re-id tasks, within a unified model. It introduces the Unbiased Prototype Consistency Learning framework, which designs an Unbiased Prototypes-guided Modality Enhancement module to enhance cross-modal shared features and multi-modal fused features simultaneously, and proposes a Cluster Prototypes Consistent Regularization module to regulate the learning of discriminative semantics across diverse object categories through prototype clustering.

**Questions:**

1. What is the reason for the large gap between fused features in the Cross-Modal Space under Distribution Divergence? It is easy to understand that in the Multi-Modal Space, the multi-modal fused features tend to be compact within the same identity due to the identity loss objective, i.e., they exhibit a small gap. A large gap in this space implies insufficient alignment across modalities. However, in the Cross-Modal Space, since the three modality-specific features are already pulled closer due to cross-modal loss and tend toward a small gap, why do the fused features still exhibit a large gap? Does simple concatenation inherently lead to a large gap?

2. Is it really necessary to unify person and vehicle re-identification tasks into a single model? Methods such as MambaPro and ICPL-ReID adopt parameter-efficient fine-tuning (PEFT) strategies, achieving strong performance with fewer trainable parameters and more flexible model designs. From this perspective, these methods can also address the problem efficiently.

3. The motivation behind cross-modal shared feature space and multi-modal fusion feature space lacks in-depth discussion.

4. Why is U_c not biased toward a dominant modality? Eq. (9) seems to be equivalent to taking the mean in Eq. (8), which is in turn based on Eq. (7). What is the difference among Eq. (7), Eq. (8), and Eq. (9)?

5. What is the performance upper bound when a single model handles different tasks? Are there more multi-modal combination experiments, such as R+N → N+T? Is there any comparison with more dataset combinations and scenarios?

6. The cross-modal retrieval performance of existing MM-ReID methods on RGBNT201, and RGBNT100 is lacking. Why does the baseline method in Table 2 outperform existing multi-modal methods that also use CLIP as the encoder? Why is there such a significant performance gap between PromptMA and DeMo?

**Ethical Concerns:**

["NO or VERY MINOR ethics concerns only"]

**Final Justification:**

The author's responses have solved my concerns.

**Limitations:**

yes

**Paper Formatting Concerns:**

Well-organized.

**Quality:**

2

**Strengths And Weaknesses:**

### Strengths:

1.This method provides a reasonable analysis of the challenges in joint training for multi-modal and cross-modal tasks, specifically focusing on learning a robust cross-modal shared feature space and an effective multi-modal fusion feature space.

2.The proposed method introduces two different types of prototypes to address modality-level and category-level multi-task problems, respectively.

3.The figures in the paper are well-drawn, and the overall presentation is relatively clear.

###  Weaknesses:
1.The motivation for integrating person and vehicle re-id into a unified model is not sufficiently justified.

2.The method shows limited innovation. The proposed Modality-Unbiased Prototypes and Category-Consistent Prototypes are essentially simple prototype aggregations, without a deep exploration of the critical role prototypes play in the M^3T-ReID task.

3.The designed contrastive loss lacks effective explanation, making it difficult to understand how it effectively addresses the practical challenges of the M^3T-ReID task.

---

> ### Author Rebuttal · Authors · 2025-07-31
>
> ## **Response to Reviewer wjAM**
> We sincerely thank you for dedicating your time to review our work and for your constructive feedback. We hope this rebuttal will help clarify our contributions and encourage a favorable reevaluation. We will also
> improve writing , clarity, correct typos, and release our code to better show the details.
>
> ---
> ### Weaknesses & Questions
>
> **W 1) & Q 2) :  Motivation and Necessity of M³T**
>
> **A1** :We thank the reviewer for the insightful comment. Real-world surveillance often involves diverse object types and multiple sensing modalities. While PEFT-based approaches like ICPL-ReID achieve competitive results with lower training costs, they are still limited to single-objective tasks and require separate models for different scenarios, resulting in redundant and expensive deployment. We propose the M³T task, which aims to address multiple retrieval modalities and identity categories within a unified model UPCL. This approach not only reduces deployment overhead but also maintains strong retrieval performance across diverse scenarios. In future work, we will explore how to leverage PEFT strategies to enhance the efficiency and adaptability of the M³T task.
>
>
> **Q 1) & Q 3):  Divergence between Cross-modal and Multi-modal Features**
>
> **A2** :Thank you for the insightful question. In the cross-modal embedding space, most existing cross-modal loss functions primarily aim to enforce modality-invariant identity representations across different modalities. However, they often neglect the preservation of modality-specific identity cues. As a result, the substantial gap observed in the fused multi-modal space does not originate from the fusion strategy itself (e.g., concatenation), but rather from the loss of modality-specific identity information during cross-modal learning, which in turn causes divergence within each modality.
>
> To further illustrate this characteristic, we conduct a feature comparison experiment between DeMO and our proposed UPCL framework. Specifically, for two image pairs of the same identity from the RGBNT201 test set, we extracted features using both DeMo and UPCL, and computed pairwise feature distances across all modalities.  The sample feature distances are respectively shown in the two tables below. Values enclosed in parentheses in the second table represent the changes relative to the corresponding figures in the first table. The results show that for the same-identity samples A and B, while the cross-modal feature distance decreases, both the intra-modal and fused multi-modal feature distances increase. This phenomenon suggests that the variation in the fused feature space stems from the loss of modality-specific identity information, rather than from the fusion strategy itself. Our method addresses identity-level and category-level consistency by mitigating the loss of modality-specific information during cross-modal learning, while simultaneously enhancing modality-specific and modality-relevant discriminative cues, thereby improving the model’s performance across diverse retrieval scenarios.
>
> | DeMO|  |    |   |  |
> | ------  | ------  | --- | --- | --- |
> |         |  RGB_B | NI_B | TI_B | RNT_B |
> |   RGB_A |   0.62  |0.88|  1.01| -|
> |   NI_A  |   0.92  |0.71|  1.16| -|
> |   TI_A  |   1.04  |1.18|  0.81|-|
> |   RNT_A  |   -  |-|  -| 0.65|
>
> | UPCL|  |    |   |  |
> | ------  | ------  | --- | --- | --- |
> |         |  RGB_B | NI_B | TI_B | RNT_B |
> |   RGB_A |   0.65(+0.03)  |0.69(-0.19)|  0.91(-0.10)|  -|
> |   NI_A  |   0.71(-0.21)  |0.73(+0.02)|  1.15(-0.01)|  -|
> |   TI_A  |   0.85(-0.19)  |1.13(-0.05)|  0.87(+0.06)| -|
> |   RNT_A  |   -  |-|  -| 0.74(+0.09)|
>
> **W 2) :  Prototype-based Innovations for the M³T Task**.
>
> **A3** :
> 1）**Conventional VI-ReID Methods in M³T**. Conventional VI-ReID cross-modal methods are not readily extendable to multi-modal retrieval scenarios and often suffer significant performance drops under more complex modality combinations. For example, we evaluated the VI-ReID-specific method DEEN on the RGBNT201 dataset. While DEEN achieves mAP/R1 scores of 22.27/22.41 and 16.51/12.77 under R-N and R-T settings respectively, its performance drastically declines to only 10.05/8.78 under the N-T setting.
>
> 2）**Multimodal ReID Methods in M³T**. Similarly, existing multi-modal methods struggle when faced with diverse retrieval categories. For instance, although the DeMO algorithm achieves strong performance on the standalone RGBNT201 dataset (79.0/82.3 in mAP/R1), its accuracy drops markedly to 64.4/63.8 under the M³T task setting, where multiple modalities and categories are jointly considered.
>
> 3）**Innovations of Our Method**.  Existing cross-modal and multi-modal methods are typically designed with fixed modality characteristics, limiting their scalability to another modalities.  Our prototype-based approach possesses greater flexibility, enabling seamless extension to a wider range of modalities and identity categories. Moreover, our prototype-based method achieves mAP/R1 scores of 22.33/23.21 (R-N), 16.77/14.59 (R-T), 17.93/14.47 (N-T), and 64.91/67.12 (multi-modal) under the M³T task setting. These results clearly surpass those of existing cross-modal and multi-modal methods. In summary, our method achieves superior experimental results within a simple framework, while demonstrating strong portability and scalability, collectively validating the innovation of our approach.
>
>
> **W 3）& Q 4) :  The Innovation and Effectiveness of Components in UPCL**
>
> **A4** :
> 1）**Discrepancies across Different Prototype Types**
> Eq. (7) aggregates features from all modalities of the same identity, but tends to be dominated by one modality. Equation (8) computes a modality-specific prototype for each individual modality, but these prototypes tend to be biased toward the characteristics of their respective modalities. To mitigate this, Equation (9) performs an aggregation over modality-specific prototypes, balancing the identity-related information from each modality. This generates a robust modality-unbiased prototype that better captures the consistent semantic representation of the corresponding identity. In addition, within CPCR, we perform clustering over modality-unbiased prototypes belonging to the same category, thereby producing in a category-consistent prototype that encapsulates class-level semantic consistency.
>
> 2）**Clarification on the Role of Contrastive Loss**.
> Based on the identity-level and the category-level consistenc semantics in the above two types of prototypes, UPCL utilizes a contrastive loss to promote more compact feature distributions for the same identity across different modalities.  Compared with the baseline, UPCL achieves average mAP/R1 improvements of 6.90/9.91 on RGBNT201 and 2.70/5.72 on RGBNT100 across all retrieval modality combinations（six cross-modal and one multimodal testing scenarios. These results clearly demonstrate the effectiveness of the proposed method.
>
>
>
>
>
>
> **Q 5) : Single-Model Performance and Extended Retrieval Scenarioss**
>
> **A5** :
> 1） **The Upper Bound of Single-model Performance**. We evaluated DEEN and DeMO on the single RGBNT201 dataset for cross-modal and multi-modal retrieval tasks, respectively. DEEN achieves mAP/R1 scores of 22.17/22.41, 16.51/12.77, and 10.05/8.78 on R-N, R-T, and N-T retrieval tasks. DeMO achieves 79.0/82.3 in the multi-modal retrieval setting. However, DeMO suffers a significant performance drop under the M³T setting, yielding 64.35/63.76（-14.65/-18.54） under the multi-modal retrieval setting. In comparison, our method achieves superior performance of 64.91/67.12, demonstrating better generalization and robustness in complex multi-modal, multi-object scenarios.
>
> 2）**Extended Retrieval Scenarioss**. Theoretically, there are 7 modality types (3 uni-modal, 3 bi-modal, 1 tri-modal), resulting in 49 total retrieval scenarios. Due to the limited number of samples in existing multi-modal ReID datasets and the lack of a standardized cross-modal evaluation metric, we focus on commonly-used retrieval setups for fair comparison. Your suggestion is highly valuable, and we plan to develop multimodal ReID datasets of new categories and explore the full spectrum of modality combinations in future work.
>
>
> **Q 6) : Analysis of Performance Differences Among Multimodal Methods**
>
> **A6** : 1) **Performance Improvement of Baseline**. Our baseline includes additional cross-modal alignment losses (all shown in Eq. 5), which directly enhance consistency across modalities. This is not present in most CLIP-based methods, hence the performance improvement.
> 2) **Comparison between DeMo and PromptMA**. DeMo designs modality-specific experts for fusion, which enhances multi-modal integration but does not explicitly optimize for cross-modal consistency. In contrast, PromptMA introduces cross-modal interactions at the prompt level, which significantly boosts cross-modal retrieval performance. Thus, the performance gap between PromptMA and DeMo reflects the difference in design focus—fusion versus alignment.

---

> ### Comment · Reviewer_wjAM · 2025-08-03
>
> Thanks for your feedback. I will revisit the authors’ response and give it further deliberation before adjusting the score.
>
> 1. Since the proposed task integrates multiple datasets, it remains unclear whether the overall training time is superior to existing single-task methods. Moreover, the use of prototypes may introduce substantial computational overhead. We encourage authors to report training time and inference efficiency compared with standard baselines.
>
> 2. Can a more generic input resolution, such as 224×224, be adopted to accommodate both pedestrians and vehicles, instead of the fixed 256×128 or 128×256 sizes?
>
> 3. A more comprehensive evaluation is needed. For example, if larger-scale datasets such as Market-1501-MM and WMVeID863 are included, will the proposed method still perform well or suffer a performance drop?

---

> > ### Author Response · Authors · 2025-08-07
> >
> > ## **Response to Reviewer wjAM**
> > Thank you once again for your time and thoughtful review of our paper. Your newly raised suggestions are truly insightful and have been very inspiring for us. We hope that our detailed responses below will satisfactorily address your concerns and lead to a favorable reconsideration of our score.
> >
> > ---
> > ### Questions
> >
> > **Q 1):  Training and Inference Efficiency**
> >
> > **A1** :
> > 1）To verify whether the overall training time across multiple datasets is superior to existing single-task methods, we conduct comparative experiments on UPCL. Specifically, UPCL takes 119 seconds per epoch when trained on the mixed dataset, while the total time for separate training is 126 seconds ( 90 seconds for RGBNT210 and 36 seconds for RGBNT100). The results show that the total training time with mixed data is more efficient than training separately.
> >
> > 2）To investigate the substantial computational overhead introduced by the incorporation of prototypes, we compare the baseline and UPCL in terms of training time, inference time, and memory usage for each epoch. The results are summarized in Table 1. We observe that the introduction of prototypes leads to an increase of approximately 9.2% in training time and 11.5% in memory usage compared to the baseline, while the inference time remains unchanged. This indicates that our method achieves significant improvements with only a marginal increase in computational overhead.
> >
> > *Table 1. **Comparison of the training time ($T_{train}$), inference time ($T_{inference}$), and memory usage ($M_{use}$) for each epoch.***
> > |         |              |                 |         |
> > | ------  | ------       | ---             | ---     |
> > |  Method |  $T_{train}$ | $T_{inference}$ | $M_{use}$|
> > | Baseline|   109s       |76s              |  16879M |
> > |   UPCL  |   119s       |76s              |  18813M |
> >
> > **Q 2):  Impact of Generic Resolution 224×224 on Model Performance**.
> >
> > **A2** :
> > To investigate whether the generic resolution of 224×224 has a significant impact on performance, we conduct comparative experiments on UPCL across different input resolutions. In these experiments, both pedestrian and vehicle images are uniformly resized to 128×256, 256×128, or 224×224. UPCL achieves mAP/Rank-1 accuracies of 21.69/23.74, 22.19/24.15, and 21.95/24.08 at the three aforementioned resolutions, respectively. Overall, compared to the 128×256 resolution we adopt, the 256×128 and 224×224 resolutions yield a negligible performance difference of around 0.50%/0.41% and 0.26%/0.34%.
> >
> > **Q 3):  Evaluation on the Mixture of Market1501MM and WMVeID863**
> >
> > **A3** :  To further validate the effectiveness of UPCL, we conduct additional experiments on Market1501MM and WMVeID863. Specifically, we construct a mixed training set by combining the training data from both datasets, and then evaluate the model separately. The following **Table 2** reports the mAP/Rank-1 accuracy obtained on each respective test set. Since DeMo is specifically designed for multimodal scenarios and lacks cross-modal supervision, it performs well in multimodal but poorly in cross-modal scenarios. Compared to DeMo, our proposed UPCL still exhibits significant superiority across various retrieval settings. These experimental results are consistent with those in our paper and validate the effectiveness of UPCL.
> >
> > We sincerely appreciate your valuable suggestion on new datasets. We plan to incorporate the experiments on the datasets you mentioned in a future revision of the paper.
> >
> > *Table 2. **Each method is trained on a mixed dataset of Market1501MM and WMVeID863, and evaluated separately. mAP(\%) / Rank-1 accuracy (\%) are reported***
> >
> > |Market1501MM|  |      |         |            |             |             |     |
> > | ------  | ------       | ---             | ---     |     ---   |  ---   | ---   | ---   |
> > |  Method       |   $R{\rightarrow}N$  | $N{\rightarrow}R$ | $R{\rightarrow}T$ |  $T{\rightarrow}R$     |   $N{\rightarrow}T$     |   $T{\rightarrow}N$     | $RNT{\rightarrow}RNT$
> > |   DeMo  | 3.12/3.05       |3.19/3.03        | 3.11/0.06 |   3.12/0.15     |    3.14/3.17    |   3.15/3.10    |   83.66/94.06    |
> > |   UPCL  |   41.17/68.76       |38.92/52.23        | 26.66/51.01 |   26.60/34.06     |    18.15/35.78    |   19.56/34.06   |   84.17/95.84    |
> >
> >
> > |WMVeID863|  |      |         |            |             |             |     |
> > | ------  | ------       | ---             | ---     |     ---   |  ---   | ---   | ---   |
> > |   Method      |   $R{\rightarrow}N$  | $N{\rightarrow}R$ | $R{\rightarrow}T$ |  $T{\rightarrow}R$     |   $N{\rightarrow}T$     |   $T{\rightarrow}N$     | $RNT{\rightarrow}RNT$
> > |   DeMo  | 3.61/4.25       |4.07/4.53        | 1.15/1.16 |   1.19/1.25    |    1.51/1.78    |   1.78/1.18    |   22.55/24.73    |
> > |   UPCL  |   11.04/12.99   |10.46/11.14        | 3.68/3.71 |   3.29/3.89    |    3.33/3.36    |   3.05/3.18    |   33.96/38.97    |

---

> > > ### Comment · Reviewer_wjAM · 2025-08-07
> > >
> > > Thanks to the authors for your efforts and replies. These responses have addressed my concerns. In my opinion, M$^3$T-ReID brings a new perspective to existing multi-modal tasks, providing strong support for solving real-world challenges. I think this deserves an increase in the final score.

---

> > > > ### Author Response · Authors · 2025-08-08
> > > > **Thank-you Letter**
> > > >
> > > > Dear Reviewer wjAM,
> > > >
> > > > We sincerely thank you for the response and improved rating. We are very grateful for your recognition of M$^3$T-ReID’s novelty and practical value in addressing multi-modal challenges. We will continue to refine our work and explore further research directions inspired by your encouraging feedback.
> > > >
> > > > All the best,
> > > >
> > > > Authors.

---

### Decision · Program_Chairs · 2025-09-17

**Decision:**

Accept (spotlight)

**Comment:**

This paper presents a new method for Multi-Modal and Multi-Task Object Re-Identification. After rebuttal and discussions, all reviewers acknowledged the novelty and strengths of the paper and agreed that the authors have addressed their concerns. Reviewers wjAM and U2xk further raised their ratings in confirming the acceptance. AC agrees with this recommendation and therefore is happy to accept the paper. Authors are required to update the rebuttal and discussion contents to the camera-ready version of the paper to improve it so as to address the raised concerns in the final paper.